# Pilot Decontamination Using Asynchronous Fractional Pilot Scheduling in Massive MIMO Systems

**DOI:** 10.3390/s20216213

**Published:** 2020-10-30

**Authors:** Muhammad Irshad Zahoor, Zheng Dou, Syed Bilal Hussain Shah, Imran Ullah Khan, Sikander Ayub, Thippa Reddy Gadekallu

**Affiliations:** 1College of Information and Communication Engineering, Harbin Engineering University, Harbin 150001, China; irshad@hrbeu.edu.cn (M.I.Z.); douzheng@hrbeu.edu.cn (Z.D.); sikander@hrbeu.edu.cn (S.A.); 2School of Software, Dalian University of Technology, Dalian 116000, China; bilalshah@dlut.edu.cn; 3College of Underwater Acoustics Engineering, Harbin Engineering University, Harbin 150001, China; 4School of Information Technology and Engineering, VIT-Vellore, Tamil Nadu 632014, India; thippareddy.g@vit.ac.in

**Keywords:** massive MIMO, pilot contamination, fractional pilot reuse, asynchronous fractional pilot scheduling

## Abstract

Due to large spectral efficiency and low power consumption, the Massive Multiple-Input-Multiple-Output (MIMO) became a promising technology for the 5G system. However, pilot contamination (PC) limits the performance of massive MIMO systems. Therefore, two pilot scheduling schemes (i.e., Fractional Pilot Reuse (FPR) and asynchronous fractional pilot scheduling scheme (AFPS)) are proposed, which significantly mitigated the PC in the uplink time division duplex (TDD) massive MIMO system. In the FPR scheme, all the users are distributed into the central cell and edge cell users depending upon their signal to interference plus noise ratio (SINR). Further, the capacity of central and edge users is derived in terms of sum-rate, and the ideal number of the pilot is calculated which significantly maximized the sum rate. In the proposed AFPS scheme, the users are grouped into central users and edge users depending upon the interference they receive. The central users are assigned the same set of pilots because these users are less affected by interference, while the edge users are assigned the orthogonal pilots because these users are severely affected by interference. Consequently, the pilot overhead is reduced and inter-cell interference (ICI) is minimized. Further, results verify that the proposed schemes outperform the previous proposed traditional schemes, in terms of improved sum rates.

## 1. Introduction

The Massive Multiple-Input-Multiple-Out (MIMO) is a promising technology for a wireless communication system that updates the traditional MIMO systems and allows the deployment of excessive antennas at the base stations (BS) of the cellular system to receive and send data at the same time [1,2,3,4]. Further, the Massive MIMO has extensive applications in Massive Internet of Things (mIoT) technology, which is used to connect billions of miscellaneous mobile devices that empowers individuals and industries to use their full potential [5,6,7] and a broad range of new applications (i.e., autonomous driving, remote health care, smart-homes, and smart-grids) are being innovated using mIoT, while the communications between massive IoT devices are controlled without human interference [8,9].

However, pilot contamination (PC) is the key issue in the development of large-scale MIMO cellular systems and limits the performance of these systems. PC occurs when a user reuses the same set of pilots in an adjacent cell or if it transmits the identical pilot in the equivalent time-frequency resource. In addition, due to PC, the problem of mutual interference occurs, which affects the accurateness of channel estimation. The mutual interference issue affects the quality of the channel estimation and makes it statistically interdependent. The authors in [10] provided an extensive survey on massive MIMO systems, pointed out several pilot contamination and computational complexity issues, and their corresponding future directions. Other than that various pilot designs (i.e., semi orthogonal pilot design and Beam domain pilot time shift) have been developed which significantly mitigated the PC issue [11,12] as shown in Figure 1. Various Blind channel and linear channel estimation schemes have been developed which successfully reduced the PC issue [4,13,14,15]. However, these systems are limited due to their computational complexity. Different pre-coding schemes have been proposed which significantly reduced the PC issue [16,17]. In addition, various pilot allocation schemes have been developed which successfully minimized the PC issue [18,19]. However, these systems are limited due to CSI overhead, and increase in complexity. Various pilot scheduling schemes have been proposed, which are based on user grouping and successfully reduced the PC issue [20,21,22]. However, these schemes are limited due to increase in pilot overhead, and become inefficient when the number of BS antennas increases.

It is indicated from above discussion that in the majority of the work proposed by authors the system performance is limited due to the increase in pilot overhead and also due to the use of power control method which becomes inefficient with the increase in the number of antennas. Moreover, many authors proposed schemes showed significant reduction in PC, but these schemes are limited due to the increase in computational complexity. Consequently, we targeted these issues and proposed two pilot scheduling schemes, i.e., Fractional Pilot Reuse (FPR) and asynchronous fractional pilot scheduling scheme (AFPS), which significantly mitigated the PC in the uplink time division duplex (TDD) massive MIMO system. Further, the pilot overhead is reduced and inter-cell interference (ICI) is minimized, in terms of improved sum rates, and the computation complexity is reduced.

The main contributions of this work are described as:

We proposed a Fractional Pilot Allocation strategy, i.e., Fractional Pilot Reuse (FPR). In the proposed FPR scheme, all the users are distributed into central and edge users according to their SINR (i.e., the value between K≤ζ≤min(3K,T)). Consequently, the optimal pilot number of the system is found which maximized the cell rate. Results verify that under certain conditions, the fractional pilot multiplexing cell rate obtained from the proposed scheme is better than traditional multiplexing schemes [23,24].We derived the capacity of central and edge users in terms of sum-rate, and proved mathematically that when the number of antennas tends to infinity the interference terms from other users with different pilot sequences can be ignored, while the pilot pollution and noise terms still exist.We proposed a user grouping pilot allocation strategy. In this strategy, we consider users in different areas of the same cell receive different levels of interference while transmitting pilots, which showed a significant reduction in pilot pollution.We proposed asynchronous fractional pilot scheduling (AFPS). In this proposed scheme, the users are grouped according to the PC they receive. The central users are assigned an identical set of pilots because of less pilot contamination they receive, while the edge users are assigned orthogonal pilots due to severe pilot contamination. Simulations results verified that the AFPS minimized the ICI, reduced the pilot overhead, and significantly reduced the PC and hence improved the overall performance of the massive MIMO system.

The remainder of the manuscript is organized as: Section 2 describes related work. In Section 3, the system model of the proposed FPR scheme is presented in detail, further the capacity of central and edge users has been derived. In Section 4 the proposed AFPS based on user grouping, is presented. The performance of our proposed schemes is compared with previous methods via simulations in Section 5. Finally, the conclusions are described in Section 6. Notations used throughout this manuscript are summarized in Table A1 in Appendix A.

## 2. Related Work

Various researchers have proposed different approaches which successfully mitigated the PC issue, as categorized via Figure 1. The literature review indicated that one way to minimize the PC is accurate pilot design methods. For instance, in [11] the authors proposed the semi-orthogonal pilot design of frequency transmission, which used the orthogonality of the asymptotic channel in the large-scale MIMO system, through continuous interference cancellation while minimizing the mutual interference between the data and pilot. This design greatly improved the performance of the MIMO system. In [12], a time-shifted pilot method with a finite number of BS antennas is proposed. The mentioned method improved the transmission performance for a small number of spatial multiplexing users. However, for a large number of multiplexing users, the scheme is ineffective.

The review indicated that linear estimation schemes or blind/semi-blind estimation schemes are also used to reduce the PC issue [4,13,14,15]. The linear estimation algorithm is used to determine the uplink (UL) channels in the time-division duplexing (TDD) protocol and obtained the downlink (DL) channel state information (CSI) by channel’s reciprocity. While using the Least-Square (LS) and Minimum-Mean-Squared-Error (MMSE) linear algorithms, the BS multiplies the received pilot signal by its conjugate transpose to obtain the desired target signal. The LS and MMSE schemes effectively reduced the impact of pilot pollution, however, in the noisy system the LS channel estimation is greatly affected, while the systems using MMSE approaches limit its performance due to its computational complexity. For instance, in [25], the authors proposed a practical maximum likelihood (ML) channel estimator and successfully mitigated the PC. The proposed estimator does not require prior knowledge of noise and interference statistics.

Various pre-coding schemes have been proposed in the literature, which successfully minimized the PC issue. For instance, in [16,17] the author designed a new large-scale fading pre-coding scheme to eliminate inter-cell interference (ICI), resulting in a considerable reduction of PC. The main idea was to linearly combine BS based on user information for users who multiplex the same set of pilot sequences. The combination coefficients depend on the fading coefficient between the user and the BS, change slowly, and are independent of the number of antennas at BS. In [14], the authors proposed a new multi-cell MMSE pre-coding method and addressed the PC issue due to the use of non-orthogonal training sequences. The authors mitigated the PC by assigning a set of training sequences to the users. Further, the authors assumed that the large-scale fading coefficients are known to all BSs. In [26], the authors proposed a polynomial extension (PE) detector for massive MIMO uplink transmission. The PE detector replaces the inverse of the pre-coding matrix by approximating the polynomial matrix.

Various pilot allocation schemes have been proposed, which successfully minimized the PC. For instance in [18,19], the authors implemented the Fractional Frequency Reuse (FFR) scheme in Long Term Evaluation (LTE) systems and successfully reduced ICI by allocating orthogonal frequency bands to the edge users in neighboring cells while using the additional spectral resources. The authors in [27] proposed a coordinated multi-point (CoMP) transmission based on frequency division duplexing (FDD) in Long Term Evaluation-Advanced (LTE-A) systems, which successfully minimized the ICI in adjacent cells. The corresponding BS receives feedback estimated by each user in the neighboring cells in the DL and distribute CSI to the neighboring cells. With the increase of BS antennas, the mentioned technique becomes inefficient because of CSI feedback overhead. To overcome the feedback overhead problem the authors in [28] proposed a non-coherent trellis-codded quantization (NTCQ) method and minimized the feedback overhead by exploiting the duality between source encoding for a moderate (32 to 64) number of antennas. However, this system is limited due to the increase in complexity with the increase in the number of BS antennas.

The authors in [20] proposed a pilot scheduling scheme based on user grouping and successfully reduced the PC as well as minimized the impact of shadow fading on the target cell. The proposed scheme assigns optimal pilots to users who greatly suffer due to PC. However, the pilot overhead increased. The authors in [21] proposed two algorithms (i.e., block diagonalization and successive optimization) and optimized the downlink pilot vectors for the multiple antennae’s users. The block diagonalization algorithm is used for throughput maximization at high signal to noise ratio (SNR), while the successive optimization algorithm improved the power control over one user time at low SNR. Both algorithms perform well if the transmit antennas are greater than the receive antennas. The authors in [22] proposed a power control method and successfully reduced the PC. This method splits coherent time into two parts and sends pilots in different time slots. However, when the number of BS antennas increases the power control method becomes inefficient. 

The authors in [29] proposed a user pilot scheduling scheme and successfully minimized the PC by estimating the mean squared error (MSE) of the users in poor channel conditions. However, although this scheme successfully reduced the PC the noise issue remains the same. The authors in [30,31] proposed a channel estimation scheme using an angle of arrival (AOA) of edge users, and successfully reduced PC. To avoid the AOA overlap, the power control method is used to reduce the interference of target central users, but the power control method becomes inefficient with the increase of BS antennas. The authors in [32] proposed a fractional pilot reuse scheme and minimized the PC. The users close to BS in adjacent cells use same pilot sequence. Further, different combining techniques are performed, to obtain the optimal number of pilots and users, which makes the system complex.

In [33], the authors proposed an FPR scheme and improved the capacity and transmission quality of the system. The scheme is divided into strict and soft FPR. Using strict FPR, edge pilots are further divided into three parts, which makes the system complex. Whereas for soft FPR, the power control method is used, which limits its performance with the increase in BS antennas. The authors in [34] proposed a pilot allocation scheme based on user grouping. In this scheme, the users are divided into the edge and central users, and pilots are allocated only to the target cell depending on the SINR they received. However, this scheme does not give any information about pilot allocation in adjacent cells. 

Literature review indicated that various pilot decontamination schemes (i.e., pilot design, channel estimation, pre-coding, and pilot allocation schemes), are limited due to the increase in pilot overhead. Further, these schemes employ the power control method thereby limiting the performance of these systems when there is an increase in the number of antennas. In addition, many proposed schemes exhibited computational complexity. Therefore, we focused on these issues and proposed FPR and AFPS schemes, which significantly mitigated the pilot overhead and inter-cell interference (ICI), resulting in a significant reduction in the PC issue, and the computation complexity, in terms of improved sum rates.

## 3. Proposed Pilot Allocation Strategy

### 3.1. System Model for Our Proposed Strategy

In our proposed scheme, an uplink of a large-scale MIMO system in a Rayleigh fading channel environment is considered, which consists of 3 hexagonal cells (i.e., S=3), and each cell is provided M BS antennas and K randomly distributed mobile users, whereas taking the condition M≫K. We assumed a Rayleigh fading channel because of the fact that Rayleigh fading channel has a positive influence on the system performance [35]. In addition, the hexagonal cell layout [36,37,38] is composed of BSs at deterministic locations, while transmitting to uniformly distributed mobile users in the cell of individual BS. For simplicity, there is one sector per cell with inner radius d, and the BS is located at the center of the cell. Each consumer is connected to the nearest BS, and hence we only took 03 hexagonal multiplexing cells from that field, in a given situation.

In our proposed scheme, the pilot scheduling is used to allocate pilots to each user in a cell according to certain rules, which significantly reduced the interference between cells. We assumed that the three users i.e., user 1-1, user 1-2, and user 1-3, use, pilot 1, pilot 2, and pilot 3 respectively. The users in adjacent cells also use the same pilot allocation. As the users, 1-1 and users 2-1 both use pilot 1 while these are very close to each other, and affects the channel estimation of BS-1 and BS-2. Hence, user 2-1 is assigned pilot 2, user 3-1 is assigned pilot 3 respectively, while user 2-2 and user 3-3 are assigned pilot 1, it is because the distance between these users is large as shown in Figure 2. While doing so the cross gain becomes large, and hence the PC problem can be greatly alleviated this way.

In our proposed strategy, the length of the orthogonal pilot used in the system is taken ζ, the pilot transmitted by the $s^{\mathrm{th}}$ cell can be expressed as $\omega_{s}=\left[\omega_{s_{1}},\omega_{s_{2}},\ldots ,\omega_{s_{K}}\right]$, where $\omega_{s_{K}}$ represents the pilot signal the $k^{\mathrm{th}}$ user transmits to the $s^{\mathrm{th}}$ cell, while the pilot signals among the cells are fully multiplexed. When cell users send pilot signals synchronously to the BS, M×ζ dimension data received by the BS can be shown as:(1)$$G_{s}=\overset{S}{\underset{j=1}{\sum }}\overset{K}{\underset{k=1}{\sum }}\sqrt{\rho_{s}\zeta }B_{\mathrm{sjk}}^{T}\omega_{\mathrm{sk}}+V_{s}$$
where $B_{\mathrm{sjk}}=[b_{s_{1}\mathrm{jk}},b_{s_{2}\mathrm{jk}},\ldots ,b_{s_{M}\mathrm{jk}}$] indicates the channel matrix, and the pilot signal satisfies as $\omega_{\mathrm{jm}}\omega_{\mathrm{in}}^{H}=\delta_{\mathrm{mn}}$. It is estimated by the LS estimation method as shows as:(2)$${\hat{b}}_{\mathrm{ssk}}=b_{\mathrm{ssk}}+\overset{S}{\underset{j=1,j\neq s}{\sum }}b_{\mathrm{sjk}}+{\hat{V}}_{s}.$$

In the above Equation (2) ${\hat{V}}_{s}=\frac{V_{s}\phi^{H}}{\sqrt{\rho_{\phi }\zeta }}\sim \mathrm{CN}\left[0,\;\frac{1}{\rho_{\phi }\zeta }I_{M}\right]$. When the BS transmits data in the downlink to the user, the signal received can be expressed as:(3)$$G_{\mathrm{sk}}=\sqrt{\rho_{d}}\overset{S}{\underset{j=1}{\sum }}\overset{K}{\underset{n=1}{\sum }}b_{\mathrm{sjn}}Q_{\mathrm{jn}}X_{\mathrm{jn}}+n_{\mathrm{sk}}.$$

As the number of antennas M at BS approaches to infinity (M→∞), then it is very convenient to use the pre-coding matrix $Q_{\mathrm{sk}}={\hat{b}}_{\mathrm{ssk}}^{H}$, and the BS transmitting power under the pre-coding is shown by $\rho_{d}\|{\hat{b}}_{\mathrm{ssk}}{\|}^{2}$, which changes with the change of the channel coherence interval. While using a constant transmit power, the BS uses a standardized pre-coding matrix as shown as:(4)$$Q_{\mathrm{sk}}=\frac{{\hat{b}}_{\mathrm{ssk}}^{H}}{\left\|{\hat{b}}_{\mathrm{ssk}}\right\|}=\frac{{\hat{b}}_{\mathrm{ssk}}^{H}}{\varphi_{\mathrm{sk}}\sqrt{M}}$$
where, $\varphi_{\mathrm{sk}}=\frac{\left\|{\hat{b}}_{\mathrm{ssk}}\right\|}{\sqrt{M}}$ represents a standardized factor which is a scalar value. Since the channel vectors from altered users to the BS are independent of each other, therefore by using theorem 1 the asymptotic expression of $\varphi_{\mathrm{sk}}{}^{2}$ can be obtained as [39,40].
(5)$$\underset{M\to \infty }{\lim}\varphi_{\mathrm{sk}}{}^{2}=\overset{S}{\underset{j=1}{\sum }}\beta_{\mathrm{sjk}}+\frac{1}{\rho_{d}\zeta }$$

If one of the received signals is extracted separately, i.e., $Z_{\mathrm{jn}}={\left|\sqrt{\rho_{d}}b_{\mathrm{ljn}}Q_{\mathrm{jn}}X_{\mathrm{jn}}\right|}^{2}$, we get the following Equation (6) as show as:(6)$$\underset{M\to \infty }{\lim}\frac{Z_{\mathrm{jn}}}{M}=\{\begin{array}{cc} \frac{\rho_{d}\beta_{\mathrm{sjn}}^{2}}{\varphi_{\mathrm{jn}}^{2}}, & n=k\\ 0, & n\neq k \end{array}$$
where $Z_{\mathrm{jn}}$ signifies the power of the signal, while the other parameters mentioned in Equation (6) are interference. Equation (2) indicates that when the number of antennas tends to infinity, the interference caused by using the same pilot does not disappear. It is because of the fact that the interference caused by the same pilot users always points towards the target users. In order to reduce the said interference, it is suggested to increase the number of orthogonal pilots, and hence fractional pilot multiplexing method can be used for this purpose. While using the said method, the orthogonal pilots can be reasonably selected and allocated to the user to avoid PC, and the corresponding reduction in interference can be achieved.

### 3.2. Fractional Pilot Multiplexing

In the previous Section 3.1, the system model of the channel follows a basic discrete-time block fading law, at certain coherent intervals while the internal channel gain matrix remains unchanged, assuming that all users simultaneously send the same time-frequency chain to the BS within a coherent interval road data [39,40]. The channel transmission coefficient of the $k^{\mathrm{th}}$ user from the $s^{\mathrm{th}}$ cell to the $m^{\mathrm{th}}$ antenna of the BS s can be expressed as:(7)$$b_{\mathrm{smjk}}=h_{\mathrm{smjk}}\sqrt{\beta_{\mathrm{sjk}}}.$$

In the above Equation (7), $h_{\mathrm{smjk}}$ indicates the fast fading coefficient, which is a cyclic symmetric complex Gaussian random variable with an independent and identical distribution of mean 0 and variance 1. The parameter $\beta_{\mathrm{sjk}}$ shows a large-scale fading factor, related to path loss and shadow fading. Since the distance between BS antennas is relatively small, hence it can be said that $\beta_{\mathrm{sjk}}$ is the same for all antennas of the same BS, and can be expressed by the following Equation (8) as:(8)$$\beta_{\mathrm{sjk}}=\frac{R_{\mathrm{ljk}}}{{\left(\frac{d_{\mathrm{ljk}}}{d_{\min}}\right)}^{\gamma }}$$
where $d_{\mathrm{ljk}}$ represents the distance between the $k^{\mathrm{th}}$ user in cell j and the central BS in cell s, the parameter $d_{\min}$ represents the user’s nearest distance from the BS in the cell, and γ represents the path loss index during signal transmission. The parameter $G_{\mathrm{sjk}}$ is the normal random variable, which is equal to $10\log_{10}\left(G_{\mathrm{sjk}}\right)$, whereas the Gaussian distribution follows the zero mean and standard deviation $\sigma_{\mathrm{shadow}}$.

In TDD channel estimation the uplink channel can be used to estimate the downlink channel, hence we only consider the uplink channel for our proposed scheme. We contemplate the performance of users in cell s, and for this, we assume the length of orthogonal pilots as ζ, and for K pilots, in each cell, the sequence can be written as $\varsigma =\left[\omega_{1},\omega_{2},\ldots ,\omega_{K}\right]\epsilon C^{\zeta \times K}$, which are orthogonal to each other. When the $s^{\mathrm{th}}$ cell user sends a pilot signal to the BS synchronously, the received pilot signal of the BS can be expressed as:(9)$$G_{s}^{\phi }=\sqrt{\rho_{\phi }\zeta }\overset{S}{\underset{j=1}{\sum }}\overset{K}{\underset{k=1}{\sum }}b_{\mathrm{sjk}}\omega_{\mathrm{jk}}^{T}+W_{s}.$$

Here, $\omega_{\mathrm{jk}}\epsilon C^{\zeta \times 1}$ characterizes the pilot sent by the $k^{\mathrm{th}}$ user in cell j, and $W_{s}\epsilon C^{M\times \zeta }$ expresses the additive white Gaussian noise (AWGN) upstream channel matrix.

The authors in [33] proposed the FPR scheme and divided FPR into strict and soft FPR schemes. This scheme improves the capacity and coverage of the system when the pilot’s SINR is high.

Unlike the strategy used in [33], in our proposed strategy, we considered 3 cells as a group (cluster). For the central users, the reuse factor 1 is taken, while taking the same set of pilot sequences; however, for the edge users, the multiplexing factor 3 is taken, and the pilot sequences are phase mutually orthogonal. Consequently, we calculated the number of central users $K_{c}$ and the edge users $K_{e}$ according to the number of cell users K and the length of pilots ζ, as shown by the following Equations (10) and (11) respectively as:(10)$$K_{c}=\frac{3K-\zeta }{2}$$
and
(11)$$K_{e}=\frac{\zeta -K}{2}.$$

The pilot sequence set ς can be divided into 4 parts according to the number of central users $K_{c}$ and edge users $K_{e}$ i.e., [$\phi_{c},\phi_{e1},\phi_{e2},\phi_{e3}$] as shown in Figure 3, which can be represented by a system model as shown in Figure 4.

In addition, the number of central users and edge users can be represented as:(12)$$\begin{array}{l} \phi_{c}=\left[\omega_{1},\omega_{2},\cdots ,\omega_{K_{c}}\right]\\ \phi_{e_{1}}=\left[\omega_{K_{c}+1},\omega_{K_{c}+2},\cdots ,\omega_{K_{c}+K_{e}}\right]\\ \phi_{e_{2}}=\left[\omega_{K_{c}+K_{e}+1},\omega_{K_{c}+K_{e}+2},\cdots ,\omega_{K_{c}+2K_{e}}\right]\\ \phi_{e_{3}}=\left[\omega_{K_{c}+2K_{e}+1},\omega_{K_{c}+2K_{e}+2},\cdots ,\omega_{\zeta }\right] \end{array}$$

The edge users and central users can be calculated according to the Equations (10) and (11), and the calculation of $K_{c}$ and $K_{e}$ is described in Algorithm 1 as shown as:


**Algorithm 1: Calculation of Central and Edge Users**
**Step 1:** Ensure that the pilots in the pilot sequence set ς are orthogonal to each other.**Step 2:** Calculate the central user $K_{c}$ and the edge user $K_{e}$ according to Equations (10) and (11).**Step 3:** The central user randomly allocates pilots in the pilot set $P_{c}.$
**Step 4:** The edge users in adjoining cells are correspondingly assigned a pilot sequence set $P_{e1},P_{e2},P_{e3}$.

### 3.3. Capacity Analysis

In this section based on Algorithm 1, the capacity of central $K_{c}$ and edge $K_{e}$ users are analyzed. We assumed a worse case, i.e., the K users in each cell send pilot sequences concurrently, and the BS uses the matched filter (MF) to receive the signal, and then analyzed the capacity of the central users as well as the edge users. At the BS of cell s the channel estimation can be shown as:(13)$${\hat{b}}_{\mathrm{ssk}}=\frac{1}{\sqrt{\rho_{\phi }\zeta }}G_{s}^{\phi }\omega_{\mathrm{sk}}.$$

The central users in altered cells reuse the same set of pilots i.e., $\omega_{\mathrm{jkc}}=\omega_{\mathrm{skc}}$, hence from Equations (9) and (13), the channel estimate of the $k_{c}^{\mathrm{th}}$ central user in cell s can be obtained as:(14)$${\hat{b}}_{{\mathrm{ssk}}_{c}}=b_{{\mathrm{ssk}}_{c}}+\underset{j\neq s}{\sum }b_{{\mathrm{ssk}}_{c}}+\frac{1}{\sqrt{\rho_{\phi }\zeta }}W_{s}.$$

The pilots in different cells used by the edge users are orthogonal to each other. Similarly, the channel estimate of the $k_{e}^{\mathrm{th}}$ edge user can be expressed as:(15)$${\hat{b}}_{{\mathrm{ssk}}_{e}}=b_{{\mathrm{ssk}}_{e}}+\frac{1}{\sqrt{\rho_{\phi }\zeta }}W_{s}.$$

Throughout the data transmission process, the user sends the data symbols, and the BS uses the estimated channel to detect the number of data symbols sent by the user, which can be shown mathematically as:(16)$$G_{s}=\sqrt{\rho_{s}}\overset{S}{\underset{j=1}{\sum }}\overset{K}{\underset{k=1}{\sum }}b_{\mathrm{sjk}}x_{\mathrm{jk}}+W_{s}$$
where, $x_{\mathrm{jk}}$ represents the data signal sent by the $k^{\mathrm{th}}$ user in cell j and the data symbol detected by the BS through the MF detector is represented by ${\hat{x}}_{\mathrm{sk}}$, hence, the $k^{\mathrm{th}}$ user data signal in the $s^{\mathrm{th}}$ cell can be expressed as:(17)$${\hat{x}}_{\mathrm{sk}}=b_{\mathrm{ssk}}^{H}G_{s}.$$

Consequently, the $k_{c}^{\mathrm{th}}$ central user data signal in the $s^{\mathrm{th}}$ cell can be represented as:(18)$${\hat{x}}_{{\mathrm{sk}}_{c}}=\sqrt{\rho_{s}}{\|b}_{{\mathrm{ssk}}_{c}}{\|}^{2}x_{{\mathrm{sk}}_{c}}+\sqrt{\rho_{s}}\overset{S}{\underset{j=1,j\neq s}{\sum }}{\|b}_{{\mathrm{ssk}}_{c}}{\|}^{2}x_{{\mathrm{jk}}_{c}}+\sqrt{\rho_{s}}\underset{\left(j1,k_{c}\right)\neq \left(j2,k\right)}{\sum }b_{{\mathrm{sj}}_{1}k_{c}}^{H}b_{{\mathrm{sj}}_{2}k}x_{j_{2}k}+W_{c}.$$

In the above Equation (18), $\sqrt{\rho_{s}}{\|b}_{{\mathrm{ssk}}_{c}}{\|}^{2}x_{{\mathrm{sk}}_{c}}$ is the expected signal,$\sqrt{\rho_{s}}\overset{S}{\underset{j=1,j\neq s}{\sum }}{\|b}_{{\mathrm{ssk}}_{c}}{\|}^{2}x_{{\mathrm{jk}}_{c}}$ is the PC, and $\sqrt{\rho_{s}}\underset{\left(j1,k_{c}\right)\neq \left(j2,k\right)}{\sum }b_{{\mathrm{sj}}_{1}k_{c}}^{H}b_{{\mathrm{sj}}_{2}k}x_{j_{2}k}+W_{c}$ is the interference along with noise $W_{c}$. Further the noise $W_{c}$ can be expressed as:(19)$$W_{c}=\overset{S}{\underset{j=1}{\sum }}b_{{\mathrm{sjk}}_{c}}^{H}n_{s}+\frac{\sqrt{\rho_{s}}}{\sqrt{\rho_{\phi }\zeta }}\overset{S}{\underset{j=1}{\sum }}\overset{K}{\underset{k=1}{\sum }}N_{s}^{H}b_{\mathrm{sjk}}x_{\mathrm{jk}}.$$

Similarly, the data signal for the edge user $K_{e}$ in a cell s can be alleviated as:(20)$${\hat{x}}_{{\mathrm{sk}}_{e}}=\sqrt{\rho_{s}}\|b_{{\mathrm{ssk}}_{e}}{\|}^{2}x_{{\mathrm{sk}}_{e}}+\sqrt{\rho_{s}}\underset{\left(j,k\right)\neq \left(1,k_{e}\right)}{\sum }b_{{\mathrm{ssk}}_{e}}^{H}b_{\mathrm{sjk}}x_{\mathrm{jk}}+W_{e}.$$

In the above Equation (20),$\;\sqrt{\rho_{s}}\|b_{{\mathrm{ssk}}_{e}}{\|}^{2}x_{{\mathrm{sk}}_{e}}$ is the expected signal of the edge user, $\sqrt{\rho_{s}}\underset{\left(j,k\right)\neq \left(1,k_{e}\right)}{\sum }b_{{\mathrm{ssk}}_{e}}^{H}b_{\mathrm{sjk}}x_{\mathrm{jk}}+W_{e}$ is the interference along with noise ($W_{e}$). In addition, the noise $W_{e}$ can be expressed as:(21)$$W_{e}=b_{{\mathrm{ssk}}_{e}}^{H}n_{s}+\frac{\sqrt{\rho_{s}}}{\sqrt{\rho_{\phi }\zeta }}\overset{S}{\underset{j=1}{\sum }}\overset{K}{\underset{k=1}{\sum }}N_{s}^{H}b_{\mathrm{sjk}}x_{\mathrm{jk}}.$$

Moreover, the capacities for central users and the edge users of the outgoing cell while using Equations (18) and (20) are derived, and are represented by the following Equations (22) and (23) respectively as:(22)$$C_{\mathrm{kc}}=E\left[\log_{2}\left\{1+\frac{\rho_{s}{\|b}_{{\mathrm{ssk}}_{c}}{\|}^{4}}{\rho_{s}\sum_{j\neq s}\|b_{{\mathrm{ssk}}_{e}}{\|}^{4}+\rho_{s}\sum_{\left(j_{1},k_{c}\right)\neq \left(j2,k\right)}{\left|b_{{\mathrm{sj}}_{1}k_{c}}^{H}b_{{\mathrm{sj}}_{2}k}\right|}^{2}+\mathrm{var}\left(W_{c}\right)}\right\}\right]$$
(23)$$C_{\mathrm{ke}}=E\left[\log_{2}\left\{1+\frac{\rho_{s}{\|b}_{{\mathrm{ssk}}_{e}}{\|}^{4}}{\rho_{s}\sum_{\left(j,k\right)\neq \left(1,k_{e}\right)}{\left|b_{{\mathrm{ssk}}_{e}}^{H}b_{\mathrm{sjk}}\right|}^{2}+\mathrm{var}\left(W_{e}\right)}\right\}\right].$$

The capacities for central and edge users are derived in Equations (22) and (23), which can further be simplified by theorem 1 as shown as:

**Theorem** **1.**
*If*
$v,u\epsilon C^{M\times 1}$
*are two independent standard complex Gaussian random vectors, and when the number of antennas*
M
*is inclined to infinity then we can have the following conditions as shown as:*
(24)$$\frac{1}{M}\left|v^{H}u\right|\to 1$$
(25)$$\frac{1}{M^{2}+2M}\|v{\|}^{4}\to 1.$$


When the number of antennas tends to infinity and $\rho_{s}=\rho_{p}=\frac{E_{s}}{M}$, the interference terms from other users with altered pilot sequences can be neglected, while the PC and noise terms still exist. Refereeing to Theorem 1, the simplified user’s capacities are shown as:(26)$$C_{\mathrm{kc}}=E\left[\log_{2}\left\{1+\frac{\rho_{s}\left(M^{2}+2M\right)\beta_{\mathrm{sskc}}^{2}}{\rho_{s}\left(M^{2}+2M\right)\sum_{j\neq s}\beta_{\mathrm{sjkc}}^{2}+M\sum_{j\neq s}\beta_{\mathrm{sjkc}}+M\frac{\rho_{s}}{\rho_{\phi }\zeta }\sum_{s=1}^{S}\sum_{k=1}^{K}\beta_{\mathrm{sjk}}}\right\}\right].$$

Similarly,
(27)$$C_{\mathrm{ke}}=E[\log_{2}\left\{1+\frac{\rho_{s}\left(M^{2}+2M\right)\beta_{\mathrm{sske}}^{2}}{M\sum_{j\neq s}\beta_{\mathrm{sjke}}+M\frac{\rho_{s}}{\rho_{\phi }\zeta }\sum_{s=1}^{S}\sum_{k=1}^{K}\beta_{\mathrm{sjk}}}\right\}].$$

From Equations (26) and (27), the total capacity of all users of cell s can be expressed as:(28)$$C_{s}=\frac{T-\zeta }{T}\overset{K}{\underset{k=1}{\sum }}C_{k}.$$

The parameter ζ characterizes the length of the pilot sequence, and T represents the coherence time, whereas the channel remains unchanged during the coherence interval.

## 4. Proposed User Grouping Pilot Allocation Strategy

In a TDD multi-cell large-scale MIMO system, using the reciprocity of the channel, the BS estimates the uplink CSI and formulate the corresponding pre-coding technology to send the downlink user information [41]. But in the channel estimation process, the channel’s coherence time frequently constrains the orthogonal pilot frequency sequences. When the number of orthogonal pilots cannot meet the number of cell users, it causes the problem of pilot pollution. In this section, we consider that users in different areas of the same cell receive different levels of interference when transmitting pilots, a solution we call user grouping is proposed to mitigate the impact of pilot pollution. In addition, in the proposed FPR pilot strategy in Section 3.2, we see that the impact of PC on mobile users in the cell is closely related to their geographical position and there is a correlation between the management locations, considering that the system reuses the identical set of pilots, if the users in adjacent cells are very close, the interference to each other increases. Similarly, if the distance between cell users increases, then there is no significant cell interference.

Hence, in this section, a user grouping pilot allocation strategy is proposed, which significantly mitigated the impact of pilot pollution. In this proposed strategy, we consider users in different areas of the same cell receiving different levels of interference while transmitting pilots. Our proposed scheme showed significant performance, in terms of reduction in the impact of pilot pollution.

Once the number of BS antennas approaches to infinity, the reachable rate of user k in cell s can be expressed as:(29)$$G_{\mathrm{sk}}=\log_{2}\left(1+\frac{\beta_{\mathrm{ssk}}^{2}}{\sum_{j\neq s}\beta_{\mathrm{sjk}}^{2}}\right).$$

Equation (29) expresses that, when the difference between direct gain and cross gain is less, the impact of PC will be very high. Moreover, if the direct gain $\beta_{\mathrm{ssk}}^{2}$ achieved is less, the SINR will be small, which affects the user’s reachable rate. In addition, the parameter $\beta_{\mathrm{sjk}}^{2}$ is difficult to obtain by the BS, while it is easier to obtain the direct gain $\beta_{\mathrm{ssk}}^{2}$. Hence, the users suffered from PC are judged based on the achieved direct gain.

Unlike the pilot scheduling as discussed earlier, in the proposed user grouping strategy in this section, the impact of the PC on the user is determined according to the cell user direct gain. If the number of users in each cell is K, then the threshold of the user group of the $s^{\mathrm{th}}$ cell can be calculated as shown as:(30)$$\rho_{s}=\frac{\lambda }{K}\overset{K}{\underset{k=1}{\sum }}\beta_{\mathrm{ssk}}^{2}.$$

Here, λ is a system parameter that can be flexibly adjusted with respect to the degree of PC in massive MIMO systems. Consequently, the users are grouped according to the threshold $\rho_{s}$ as shown as:(31)$$\beta_{\mathrm{ssk}}^{2}>\rho_{s}\;\{\begin{array}{cc} \mathrm{Established} & \mathrm{Central}\;\mathrm{user}\\ \mathrm{not}\;\mathrm{Established} & \mathrm{Edge}\;\mathrm{user} \end{array}.$$

As the central user is far away from the users in other cells, and the interference from users in other cells is relatively small, hence we call this a central user group. Similarly, the users at the edge of the cell are close to the edge user and are highly affected by interference from users in other cells, hence we call this an edge user group.

Unlike the pilot allocation scheme proposed in [42], in our proposed scheme, the cell users are grouped as derived in Equations (30) and (31). Further, the pilots are allocated to central and edge users as shown in Algorithm 2.


**Algorithm 2: Pilot Allocation to Central and Edge Users**
**Step 1:** Randomly select F neighboring cells from the system, and record the target cell as s, the number of cell users is K, and the pilot number is recorded as ζ, and the large-scale fading factor is determined from the user to the BS.**Step 2:** Choose the appropriate parameter λ, calculate the grouping threshold using equation (28), and record $K_{c}=0,{\;K}_{e}=0$.**Step 3:** Check the cell users one by one and compare it with the threshold. If the user is greater than the given threshold, then $K_{c}=K_{c}+1,$ otherwise $K_{e}=K_{e}+1$.**Step 4:** Number of all orthogonal pilots are $\Omega_{1,\;}\Omega_{2,},\ldots ,\Omega_{\zeta ,}$. First, the central users are assigned the value from $\Omega_{1}$ to $\Omega_{k_{c}},$ then assign pilots to the remaining $K_{e}$ users. For $K_{c}$ users in other cells assign the same value from $\Omega_{1}$ to $\Omega_{k_{c}}$, and the edge user continues to allocate orthogonal pilots.**Step 5:** Other cells of the system can be allocated following the above steps.

The specific allocation strategy as described in Algorithm 2, is shown in Figure 5. The users with low PC in the circle are shown by the same color indicating the same group of pilots allocated, while orthogonal pilots are allocated to the users which are outside of the circle.

### Asynchronous Fractional Pilot Scheduling Scheme for Central Users

In this section, we proposed an asynchronous scheme for central users, which significantly reduced the PC issue while sending the uplink pilot. In this proposed strategy, when the central user of the adjacent cells sends the same set of pilots in different time slots, in the meanwhile all the cells of BSs send downlink data to users located at the edge of the cell. Similarly, when all the central users in the cell send the pilots, in the meantime the edge users in apiece cells start sending orthogonal pilots, while the BS sends the downlink data to the central users of the cell. In addition, the base station can distinguish the uplink pilot signal received from another central user and the downlink data sent by itself. Once, the edge user sends the pilot, then at the same time all cell users send uplink data to the corresponding BS, while considerably eliminating the problem of PC in the central users of the cell.

We consider the TDD mode of large-scale MIMO systems for transmission. We assume that the channel response remains unchanged in a coherent time, when the central user multiplexes the same set of pilots while avoiding the overlapping in the time domain, resulting in a significant reduction in the PC. Further, the number of users in the target cell s can be expressed as:(32)$$K_{s}=K_{s,c}+K_{s,e}.$$
where, $K_{s,c}$ represents the number of center users in target cell $s,\;K_{s,e}$ indicates the number of edge users in target cell s. Hence the number of pilots used can be expressed as:(33)$$K_{p}=K_{c}+K_{e}$$
where $K_{c}$ represents the number of pilots assigned to the central user, and, $K_{e}=$
$\overset{S}{\underset{s=1}{\sum }}K_{s,e}$ represents the number of pilots allotted to the edge user. If the pilot set is recorded as $\vartheta \epsilon C^{\zeta \times K}$ then it can be divided as shown as:(34)$$\vartheta =\vartheta_{c}\vartheta_{e}.$$

In the ideal scenario, while the pilot is sent asynchronously, the duration of the uplink pilot signal sent by all users in the cell at a similar time is shown as $T_{p}$. The total pilot transmission time is split into L small time segments, and the central users in different cells send uplink pilot signals in non-overlapping time slots, which can be expressed as $T_{1,p,c},T_{2,p,c},T_{3,p,c},\ldots ,T_{L,p,c}$. Consequently, the PC issue is reduced to a minimum, as shown by the following Equation (35) as:(35)$$\overset{S}{\underset{j=1,j\neq s}{\sum }}\overset{K}{\underset{k=1}{\sum }}\Omega_{j,k,c}\Omega_{s,k,c}^{T}=0.$$

In addition, the mutually orthogonal pilots are assigned to $\Omega_{j,k,c}\epsilon \vartheta_{c},{\;\Omega }_{s,k,c}\epsilon \vartheta_{c}$ as well as the edge users are assigned mutually orthogonal pilots, as shown in Figure 6:

When the central user sends pilots at different time slots, the pilot received by BS s can be expressed as:(36)$$G_{s,p}=\sqrt{\rho_{\phi }\zeta }\overset{S}{\underset{j=1}{\sum }}\overset{K}{\underset{k=1}{\sum }}b_{\mathrm{sjk}}^{T}\Omega_{s,k}^{T}+W_{s}=\sqrt{\rho_{\phi }\zeta }\overset{K_{s,c}}{\underset{k=1}{\sum }}b_{\mathrm{ssk}}\Omega_{s,k,c}^{T}+\sqrt{\rho_{\phi }\zeta }\overset{K_{s,e}}{\underset{k=1}{\sum }}b_{\mathrm{ssk}}\Omega_{s,k,e}^{T}+\sqrt{\rho_{\phi }\zeta }\overset{S}{\underset{j=1,j\neq s}{\sum }}\overset{K_{s,e}}{\underset{k=1}{\sum }}b_{\mathrm{sjk}}^{T}\Omega_{s,k,e}^{T}+W_{s}$$
where, $W_{s}$ is AWGN independently and identically distributed. When the $s^{\mathrm{th}}$ cell BS is detected then it is used to estimate the channel as shown as:(37)$${\hat{b}}_{\mathrm{ssk},c}=\frac{1}{\sqrt{\rho_{\phi }\zeta }}{G\Omega }_{s,k,c}^{*}=b_{\mathrm{ssk},c}+\frac{1}{\sqrt{\rho_{\phi }\zeta }}W_{s}\Omega_{s,k,c}^{*}=s_{\mathrm{ssk},c}+\Omega_{s,k,c}.$$

Meanwhile, the channel estimation of the edge users of the cell acquired by the BS can be expressed as:(38)$${\hat{b}}_{\mathrm{ssk},e}=\frac{1}{\sqrt{\rho_{\phi }\zeta }}G_{s,p}\Omega_{s,k,e}^{*}=b_{\mathrm{ssk},e}+\frac{1}{\sqrt{\rho_{\phi }\zeta }}W_{s}\Omega_{s,k,e}^{*}=b_{\mathrm{ssk},e}+\Omega_{s,k,e}$$
where, $\Omega_{s,k,c}$ and $\Omega_{s,k,e}$ are noise terms, and does not affect the pilot frequency. After the pilot transmission, the uplink data signal received by BS s can be expressed as:(39)$$G_{s}=\sqrt{\rho_{u}\zeta }\overset{S}{\underset{j=1}{\sum }}\overset{K}{\underset{k=1}{\sum }}b_{\mathrm{sjk}}x_{\mathrm{jk}}+W_{s}.$$

Here, the MF detector is employed to detect uplink data transmission based on channel estimation, and hence the $k^{\mathrm{th}}$ central user in s cells can be expressed as:(40)$${\hat{x}}_{\mathrm{sk}.c}={\left({\hat{b}}_{\mathrm{ssk},c}\right)}^{H}G_{s}={\left(b_{\mathrm{ssk},c}+\Omega_{s,k,c}\right)}^{H}\left\{\sqrt{\rho_{s}\zeta }\overset{S}{\underset{j=1}{\sum }}\overset{K}{\underset{k=1}{\sum }}b_{\mathrm{sjk}}x_{\mathrm{jk}}+W_{s}\right\}=\sqrt{\rho_{s}\zeta }\left(b_{\mathrm{ssk},c}^{H}b_{\mathrm{ssk},c}x_{\mathrm{sk}}\right)+\xi_{\mathrm{sk}.c}$$

Whereas, the parameter $\xi_{\mathrm{sk}.c}$ represent the incoherent term and noise, which minimize gradually with the increase in the number of BS antennas. Hence, the SINR of the central user k in the $s^{\mathrm{th}}$ cell can be stated as:(41)$${\mathrm{SIN}R}_{\mathrm{sk}.c}=\frac{{\left|b_{\mathrm{ssk},c}^{H}b_{\mathrm{ssk},c}\right|}^{2}}{\frac{{\left|\xi_{\mathrm{sk}.c}\right|}^{2}}{\rho_{s}}}.$$

Similarly, the $k^{\mathrm{th}}$ edge user in $s^{\mathrm{th}}$ cell can be expressed by the following Equation (42) as:(42)$${\hat{x}}_{\mathrm{sk}.e}={\left({\hat{b}}_{\mathrm{ssk},e}\right)}^{H}G_{s}={\left(b_{\mathrm{ssk},e}+\Omega_{s,k,e}\right)}^{H}\left\{\sqrt{\rho_{s}\zeta }\overset{S}{\underset{j=1}{\sum }}\overset{K}{\underset{k=1}{\sum }}b_{\mathrm{sjk}}x_{\mathrm{jk}}+W_{e}\right\}.$$

The parameters $\xi_{\mathrm{sk}.c}$ and $\xi_{\mathrm{sk}.e}$ are similar, indicating incoherent terms and noise. Hence, the SINR of the edge user k in $s^{\mathrm{th}}$ cell can be expressed as:(43)$${\mathrm{SIN}R}_{\mathrm{sk}.e}=\frac{{\left|b_{\mathrm{ssk},e}^{H}b_{\mathrm{ssk},e}\right|}^{2}}{\frac{{\left|\xi_{\mathrm{sk}.e}\right|}^{2}}{\rho_{s}}}.$$

Therefore the average achievable rate of the user k in cell s can be represented as:(44)$$G_{\mathrm{sk}}=E\left\{P\left(\beta_{\mathrm{ssk}}^{2}>P_{s}\right)\log_{2}\left(1+{\mathrm{SIN}R}_{\mathrm{sk}.c}\right)+\left(1-P\left(\beta_{\mathrm{ssk}}^{2}>P_{s}\right)\right)\log_{2}\left(1+{\mathrm{SIN}R}_{\mathrm{sk}.c}\right)\right\}.$$

Whereas the parameter P represents the possibility that the user is the central user, hence the achievable sum rate of cell s can be expressed as:(45)$$C_{s}=\frac{T-\zeta }{T}\overset{K}{\underset{k=1}{\sum }}G_{\mathrm{sk}}.$$

Here, the parameter ζ indicates the length of the pilot sequence, and T expresses the channel coherence interval, while the channel transmission coefficient remains unchanged within the channel coherence interval.

## 5. Results and Discussion

In this section, we demonstrated the performance of the proposed fractional pilot multiplexing strategy by using simulation software. Regarding this, some important parameters used in the simulation process are given in Table 1. We assumed that users in the system are evenly distributed in each cell, and the BS is located in the center of the cell. The cellular network is composed of three cells i.e., (S=3), and the radius of each cell is taken d=1000 m. The least distance from the user to the BS is $d_{\min}$ = 100 m, because $\beta_{\mathrm{sjk}}$ is inversely proportional to $d_{\mathrm{sjk}}$. As a result, we only need to consider the adjacent cell for the interference and noise, the path loss index γ = 3.8 during the signal transmission, and the standard deviation of shadow fading σ = 8 dB. In order to sustain the generality, it is assumed that every 3 cells form a multiplexing unit, and the length of each orthogonal pilot is the same as the number of orthogonal pilots used by the system.

Figure 7 shows the relationship between the sum rate and the number of pilots. The number of BS antennas taken is 64 i.e., (M=64), the number of users taken is 15 i.e., (K=15), and coherence time is T=196 s.

In the proposed allocation strategy, the pilot number is adaptive according to different SNR, and its value is between K≤ζ≤min(3K, T). Considering this limited case, when the multiplexing factor is 1, the number of pilots is K, and when the multiplexing factor is 3, the number of pilot is 3K. From Figure 7, it can be observed that the cell rate rises first and then falls as the number of pilots escalates, increasing the number of orthogonal pilots due to which the channel estimation error becomes smaller and consequently the user reachable rate increases.

However, when the number of pilots increases to a certain extent, the channel estimation becomes more accurate, and the data transmission time decreases while the reachable rate gradually decreases. Hence, there is a balance point between the pilot and signal transmission, which maximizes the sum rate of the cells. To increase the system capacity, the optimal number of pilot reuse is calculated while dividing the number of pilots by the number of cells in the multiplexing unit. With the optimal number of pilot reuse, the proposed strategy has a higher system capacity than the traditional scheme in [23,24].

Figure 8 shows the effect of different SNR on the number of orthogonal pilots used by the cell. When the SNR is small, the increase in the number of pilots is not obvious, however, when the SNR reaches to 20 dB, the number of pilot’s increases rapidly to the maximum. After a further increase in SNR (i.e., more than 20 dB), the influence of PC on the system performance becomes progressively obvious, in terms of reduction in PC due to an increase in the number of orthogonal pilot sequences. When the SNR further increases to 28 dB, the system uses the pilot multiplexing factor of 3 for the edge users as described in Algorithm 1, and the adjacent cells also use the orthogonal pilot sequence to overcome the pilot pollution. Which indicated a significant impact in terms of eliminating the PC.

Figure 9 presents the effect on sum-rate with the different pilot reuse factors at the BS. The central users we assumed are eight i.e., ($K_{c}$ = 8) and the edge users are seven i.e., ($K_{e}$ = 7), and hence the reuse factor achieved is 1.9. It can be seen that as the number of antennas increases, the sum rate increases gradually. It can be observed from Figure 9 that, the fractional pilot reuse is better with a factor of 1 and 3. In addition, the MF filters are employed at the BS to receive orthogonal signals, if the pilot reuse factor 3 is used, and hence significant performance improvement is achieved consequently.

Figure 10 presents the effect on the sum-rate with a different number of BS antennas. Several common classical fractional pilot multiplexing algorithms are compared with our proposed strategy. In our proposed strategy, the central $K_{c}$ and edge $K_{e}$ users based on the total number of users K and pilot sequence ζ, are calculated first and then assigned the pilots to edge users as we defined in Algorithm 1. From Figure 10, it can be observed that with the increase in the number of BS antennas the sum rate increases significantly. Further, the proposed scheme showed significant advantages in terms of an increase in the sum rate as compared to soft pilot reuse (SPR) [20] and the conventional algorithm [33].

We also evaluated the performance of the proposed AFPS scheme by using simulation. We used Monte Carlo simulations in order to evaluate the performance of the proposed scheme in a multi-cell large-scale MIMO system environment. The cellular network we used consists of three cells i.e., (S=3), the cell radius is taken d=1000 m, and the least length from the user to the BS is taken $d_{\min}=100$ m, while the path loss index taken is γ = 3.8 during signal transmission, and the standard deviation of shadow fading taken is σ = 8 dB. We assumed that this system has no power distribution, and all the users in the cell have the same power to transmit pilot and data to the BS with power $\rho_{p}=\rho_{s}=10\;\mathrm{dB}$. We assumed that every 3 cells form a multiplexing unit i.e., (F = 3), while the length of a single orthogonal pilot is the same as the number of orthogonal pilots used by the system.

Figure 11 shows the relationship between the uplink reachable rate and the number of BS antennas with different pilot scheduling schemes. From Figure 11, it can be observed that the performance of the proposed scheme is significantly better than the SPR scheme in [20] and the traditional schemes in [23]. Figure 11 depicts that when the number of antennas, i.e., M is small, the reachable rate of the proposed scheme is slightly higher than the traditional schemes, but when the number of antennas approaches to 128 (i.e., M=128) the sum rate of the proposed scheme is much higher than that of the previous schemes [24], and the cell throughput of the proposed scheme is about 3 bps/Hz higher than that of asynchronous pilot scheduling (APS) scheme [34], which is mainly because the central user has no PC. In addition, when the number of antennas approaches to 512 (i.e., M=512), the cell throughput difference of the proposed and SPR scheme is about 15–20 bps/Hz.

Figure 12 shows the effect of separation parameter λ on the uplink rate, with a different interval of channel coherence time T. We assumed the number of BS antennas 128 (i.e., M=128), and cell users 15 (i.e., K=15). The parameter λ depends on the other system parameters, such as the number of users K and the number of BS antennas M, whereas the parameter λ has a substantial impact on system performance, hence it is selected carefully to obtain better performance of the system. When the interval of channel coherence is small, i.e., T=100, we select λ(λ>3), which minimizes the number of orthogonal pilot sequences and maximizes the resources allocated for data transmission. Further, when the interval of channel coherence is large, i.e., T=300, the smaller value of λ is selected, i.e., 1.5<λ<3. For the medium channel coherence interval, the value is λ≈2.

Figure 13 illustrates the effect of channel coherence time on the sum rate. We assumed the number of BS antennas to 128, the cell users 15, and the separation parameter is taken as λ=1.9. Figure 13 depicts that, as the coherence time increases, the cell reachable rate also increases and consequently improves the sum rate of the cell. Figure 13 depicts that, the proposed scheme and APS scheme showed a significant increase in sum rate as compared to SPR in [20] and the traditional schemes in [23]. It is due to the fact that in our proposed scheme, the pilot distribution is performed after the grouping of the users. The central users in adjacent cells transmit the identical set of pilot sequence in different time slots in the uplink, and the BSs of all cells transmits downlink data to the edge users. Further, when the edge users in all cells send orthogonal pilots in the uplink, the BSs send downlink data to the central users, and hence effectively eliminated the PC. Additionally, it can be observed from Figure 13 that, as the coherence time is increased to T=1000, the sum rate of the proposed scheme is 3 bps/Hz higher than the APS scheme [34] and showed significant mitigation in the PC.

## 6. Conclusions

In this article, we proposed fractional pilot reuse and asynchronous fractional pilot scheduling schemes which improved the sum rate as well as eliminated the PC in the TDD massive MIMO system. We employed the proposed FPR scheme in a massive MIMO system and calculated the optimal number of pilots which improved the system performance and increased the capacity of the system. Further, the proposed AFPS scheme effectively eliminated PC and minimized the ICI. Results verify that the proposed schemes effectively reduced the pilot overhead and enhanced system performance. Furthermore, the sum rate of the system is improved significantly.

Our proposed schemes performance becomes worse when the number of users in the cell increases, and hence further research work is needed to coup this issue and to improve the system performance in the future.

## Figures and Tables

**Figure 1 sensors-20-06213-f001:**
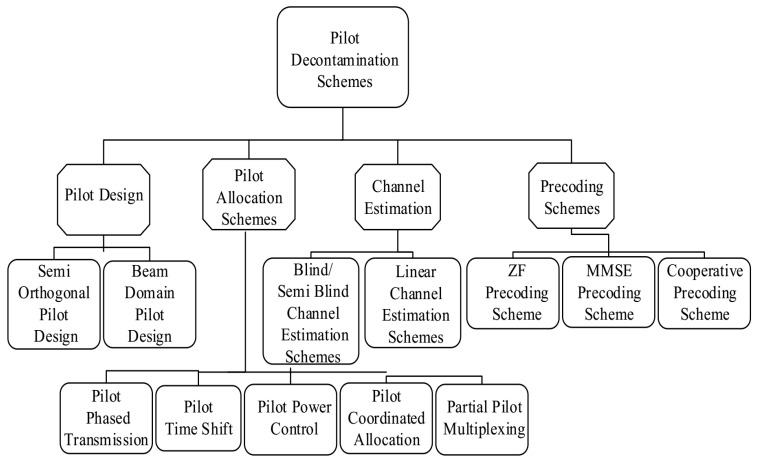
Research status of pilot contamination mitigation from four aspects: pilot design schemes, pilot allocation schemes, channel estimation schemes, and pre-coding schemes.

**Figure 2 sensors-20-06213-f002:**
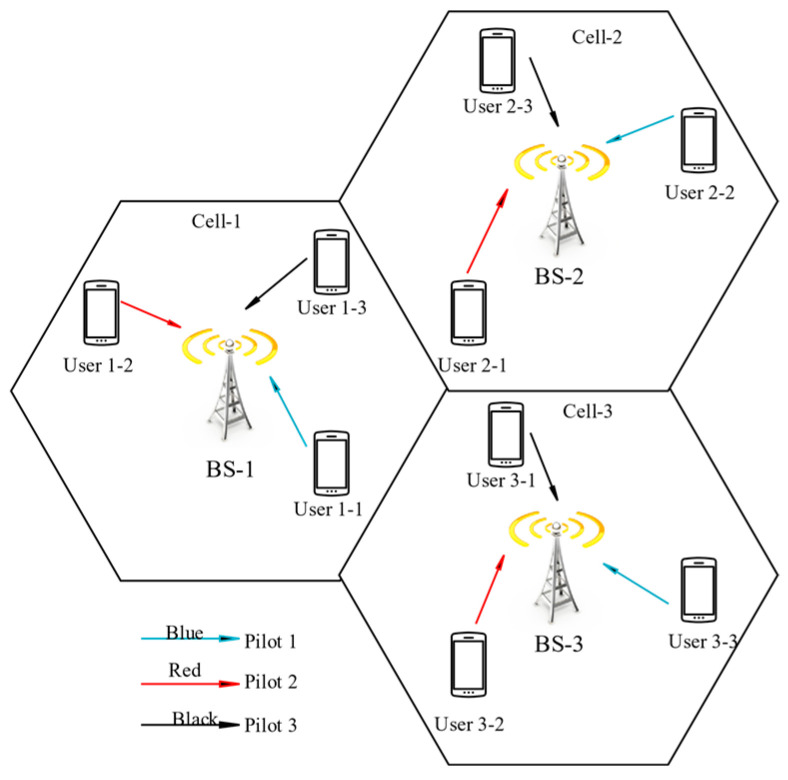
Pilot scheduling system model schematic diagram.

**Figure 3 sensors-20-06213-f003:**
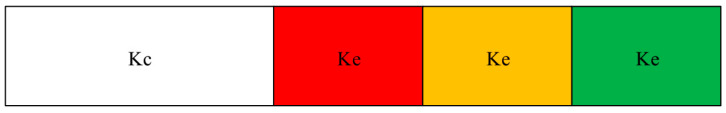
Pilot sequence set.

**Figure 4 sensors-20-06213-f004:**
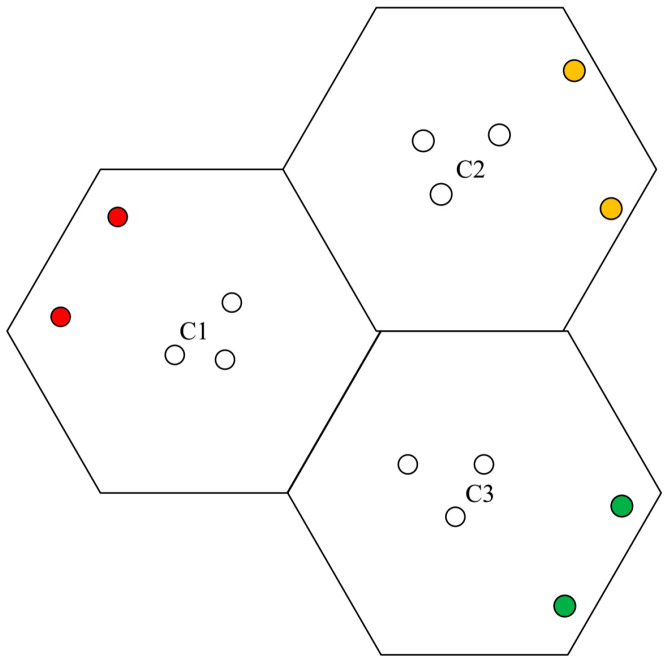
Fractional pilot multiplexing system model.

**Figure 5 sensors-20-06213-f005:**
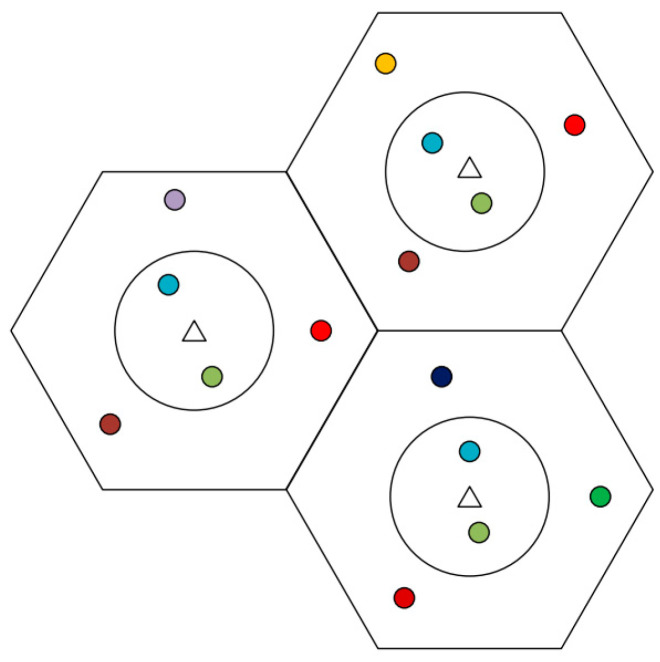
The system model of pilot allocation.

**Figure 6 sensors-20-06213-f006:**
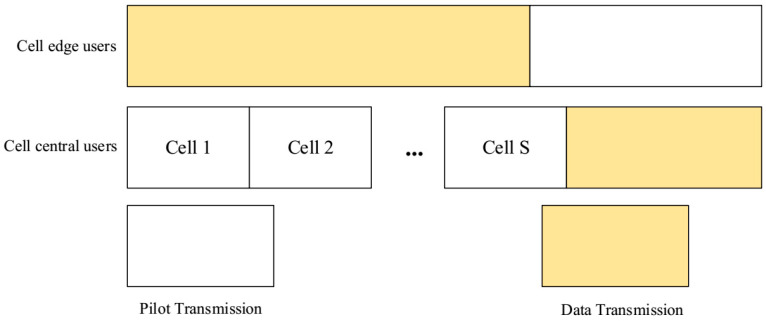
Schematic diagram of the asynchronous fractional pilot transmission.

**Figure 7 sensors-20-06213-f007:**
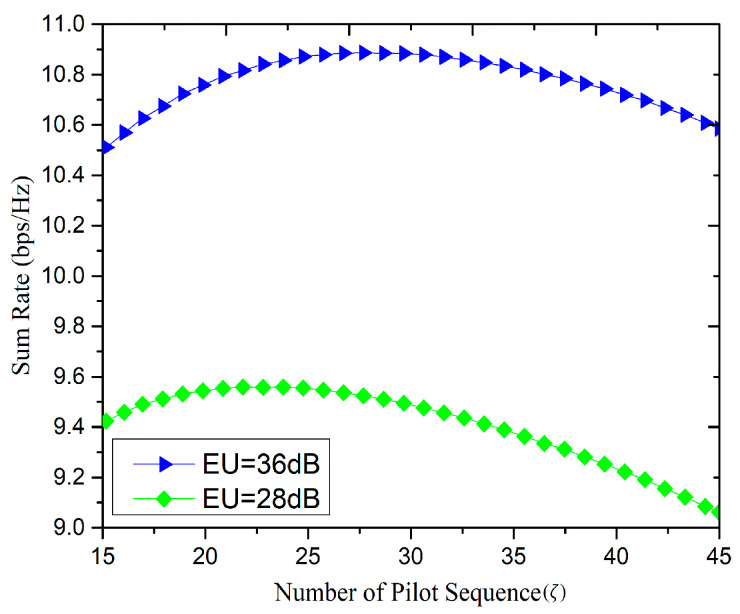
Relationship between sum-rate and number of pilot sequence (ζ).

**Figure 8 sensors-20-06213-f008:**
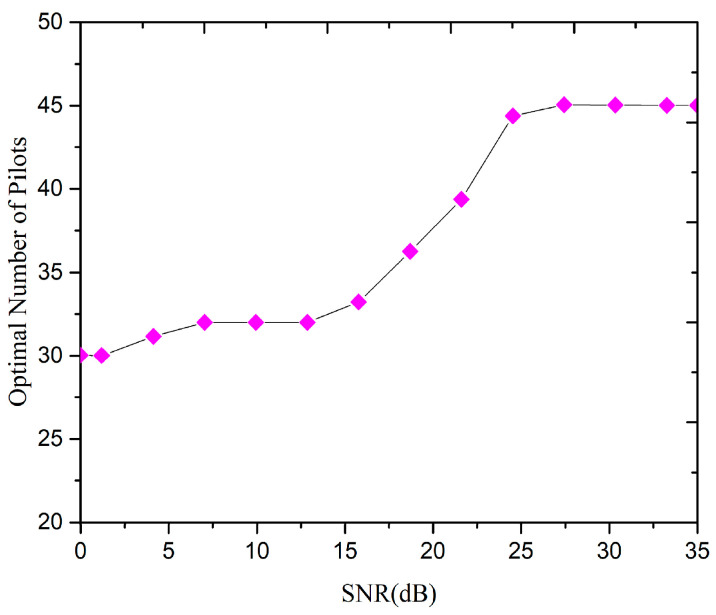
Relationship between optimal number of pilots and signal to noise ratio (SNR).

**Figure 9 sensors-20-06213-f009:**
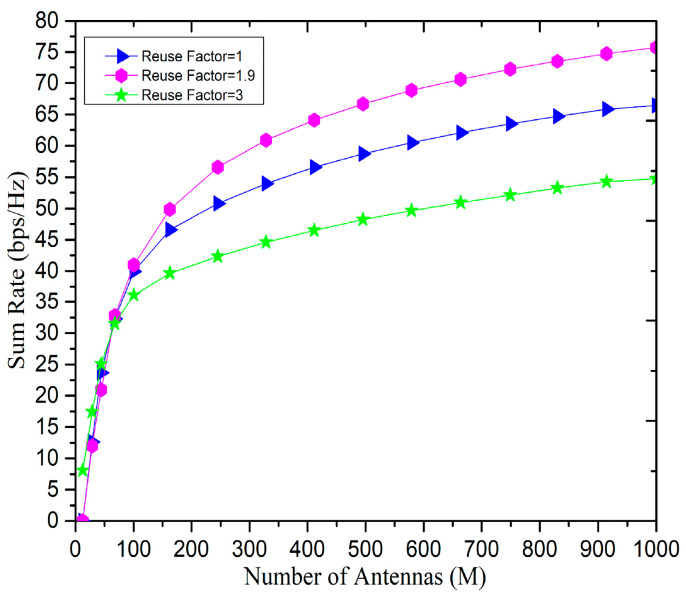
The variation in sum-rate with the number of antennas (M) under the proposed pilot multiplexing scheme.

**Figure 10 sensors-20-06213-f010:**
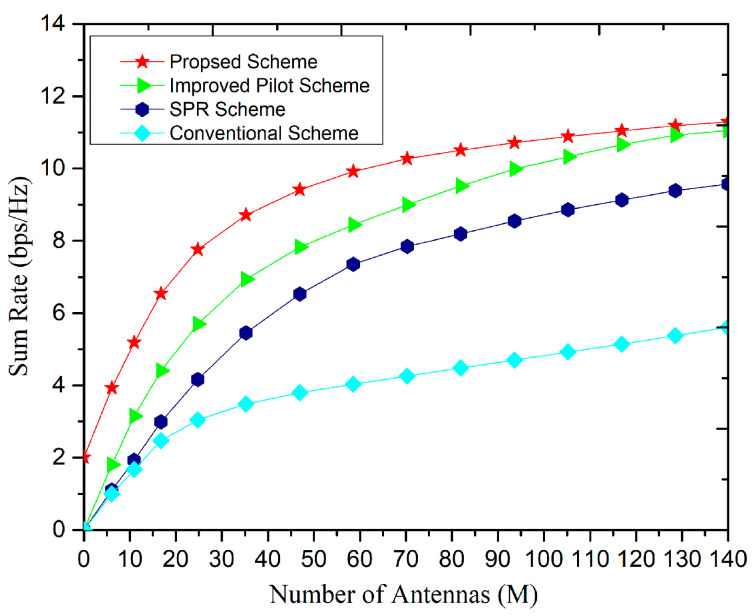
Variation of the sum-rate with the number of antennas (M) under different schemes.

**Figure 11 sensors-20-06213-f011:**
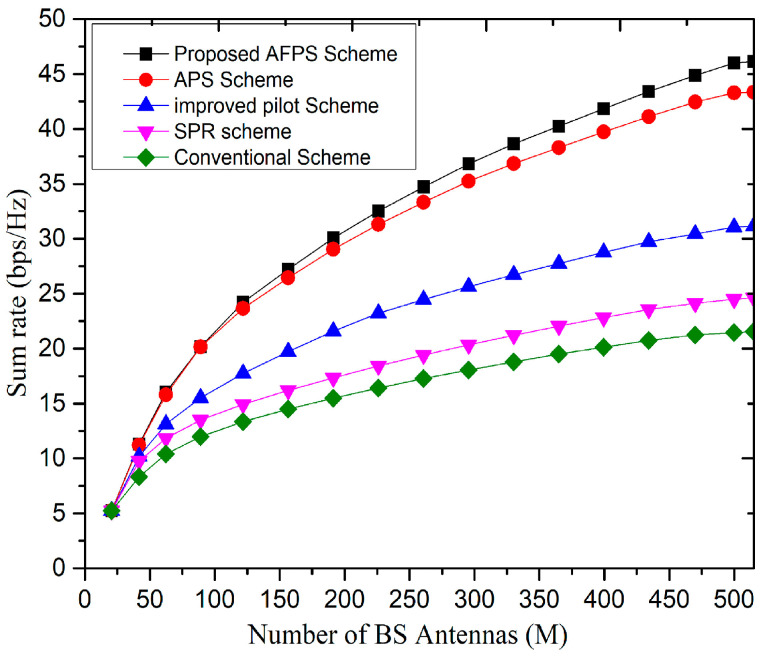
Relationship between sum-rate and number of antennas (M).

**Figure 12 sensors-20-06213-f012:**
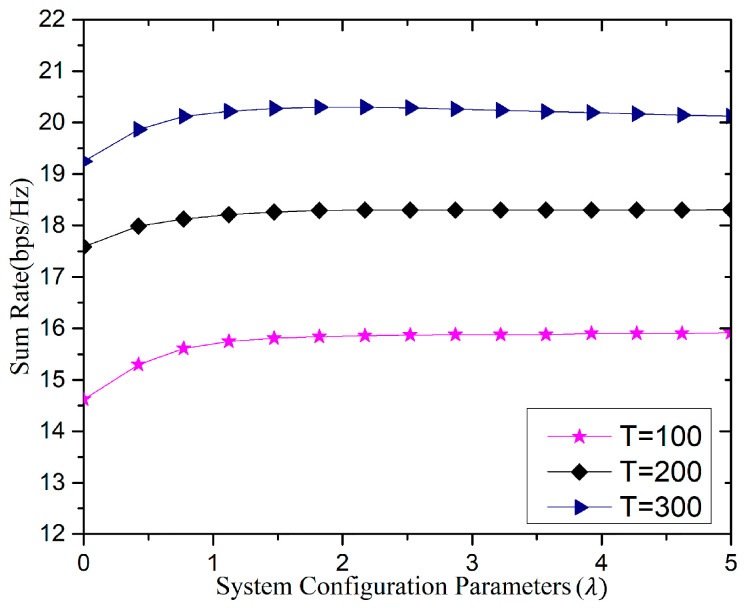
Relationship between sum-rate and system configuration parameter (λ).

**Figure 13 sensors-20-06213-f013:**
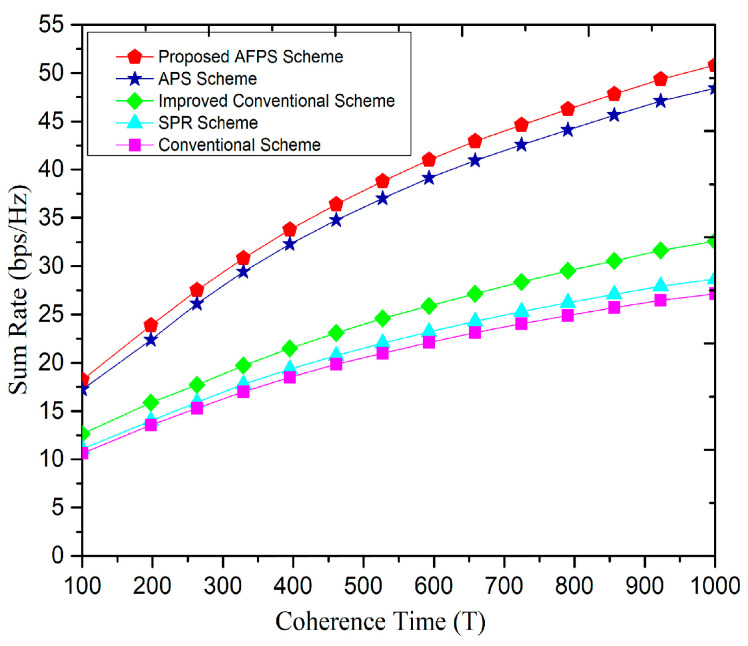
Relationship between sum-rate and coherence time.

**Table 1 sensors-20-06213-t001:** Simulation parameters.

S/NO.	Parameters	Values
1	Radius (d)	1000 m
2	$d_{\min}$	100 m
3	Path loss index (γ)	3.8
4	Number of cells S	3
5	shadow fading (σ)	8 dB
6	$\rho_{s}$	10 dB
7	F	3
8	Number of users (K)	15
9	Number of central users ($K_{c}$)	8
10	Number of edge users ($K_{e}$)	7
11	Pilot length (ζ)	4
12	Channel	Rayleigh Channel

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
