# Peer review of "Pilot Decontamination Using Asynchronous Fractional Pilot Scheduling in Massive MIMO Systems"

_sensors, 2020, doi:10.3390/s20216213_

Round 1
Reviewer 1 Report
This is a very interesting article on pilot decontamination in massive MIMO systems where the authors have used asynchronous fractional pilot scheduling to mitigate the problem, however, this reviewer has some suggestions for improving the text's quality
1) The introduction should be improved, you should sell your novelty better. Try to create some bullets points for each one of the novelties you propose.
2) It is highly recommended to add a Literature review section, where you should contrast your work with the works of other researchers on the same area or topic.
3) Improve the number of references. If you add a proper literature review section it will be straightforward to have more references. Below is a short list of papers you could add to your literature review and consequently to your references:
[1] Olakunle Elijah, Chee Yen Leow, Tharek Abdul Rahman, Solomon Nunoo, and Solomon Zakwoi Iliya, A Comprehensive Survey of Pilot Contamination in Massive MIMO-5G System, IEEE Communications Surveys & Tutorials, vol. 18, no. 2, pp. 905-923, 2016.
[2] Felipe A. P. de Figueiredo, Fabbryccio A. C. M. Cardoso, Ingrid Moerman, and Gustavo Fraidenraich, Channel estimation for massive MIMO TDD systems assuming pilot contamination and flat fading, EURASIP Journal on Wireless Communications and Networking, vol. 2018, no. 14, January, 2018.
[3] J. Jose, A. Ashikhmin, T. L. Marzetta, and S. Vishwanath, Pilot contamination and precoding in multi-cell TDD systems, IEEE Trans. Wireless Commun., vol. 10, no. 8, pp. 2640-2651, Aug. 2011.
3) All figures should have their quality improved. All of them seem to be blurred.Please, when generating the figures, generate a pdf figure and add that .pdf to your paper instead of the current figures.
Author Response
Response to Reviewer 1 Comments
This is a very interesting article on pilot decontamination in massive MIMO systems where the authors have used asynchronous fractional pilot scheduling to mitigate the problem, however, this reviewer has some suggestions for improving the text's quality
- The introduction should be improved, you should sell your novelty better. Try to create some bullets points for each one of the novelties you propose.
Answer: As instructed by respected reviewers, the introduction section is updated and improved, and highlighted the novelty of our work, and presented the novelty of our work by some bullets points.
- It is highly recommended to add a Literature review section, where you should contrast your work with the works of other researchers on the same area or topic.
Answer: The literature review section (i.e. section 2) has been included in the manuscript, representing the contrast of our work with the works of other researchers on the pilot contamination (PC) issue.
- Improve the number of references. If you add a proper literature review section it will be straightforward to have more references. Below is a short list of papers you could add to your literature review and consequently to your references:
[1] Olakunle Elijah, Chee Yen Leow, Tharek Abdul Rahman, Solomon Nunoo, and Solomon Zakwoi Iliya, A Comprehensive Survey of Pilot Contamination in Massive MIMO-5G System, IEEE Communications Surveys & Tutorials, vol. 18, no. 2, pp. 905-923, 2016.
[2] Felipe A. P. de Figueiredo, Fabbryccio A. C. M. Cardoso, Ingrid Moerman, and Gustavo Fraidenraich, Channel estimation for massive MIMO TDD systems assuming pilot contamination and flat fading, EURASIP Journal on Wireless Communications and Networking, vol. 2018, no. 14, January 2018.
[3] J. Jose, A. Ashikhmin, T. L. Marzetta, and S. Vishwanath, Pilot contamination and precoding in multi-cell TDD systems, IEEE Trans. Wireless Commun., vol. 10, no. 8, pp. 2640-2651, Aug. 2011.
Answer: As suggested by the esteemed reviewer, we have added the recommended references in the related work section (section 2) in the manuscript and cited correspondingly in the manuscript.
- All figures should have their quality improved. All of them seem to be blurred. Please, when generating the figures, generate a pdf figure and add that .pdf to your paper instead of the current figures.
Answer: According to reviewer’s comments, all the figures are re-generated in the pdf files, and also improved their quality and added in the manuscript accordingly.
- Proofread the manuscript looking for grammar errors and typos. During my reading, I found some errors that should be corrected after a careful reading.
Answer: The manuscript has been carefully read by the authors and improved all the grammatical errors and typos.

Reviewer 2 Report
Overall, this is a nice contribution on the topic of pilot decontamination in massive MIMO systems. Still, before publication, it is my opinion that the following major comments should be addressed by the authors:
1) Please define all the acronyms in their first use, e.g. TDD.
2) Sec. I – “Various researchers have proposed different approaches and mitigated the issue of PC, which are shown in Figure 1” -> “Various researchers have proposed different approaches to mitigate the issue of PC, as categorized via Figure 1”
3) When introducing the benefits of massive MIMO technology (at the beginning of Sec. I), the authors may also want to highlight its benefit to some important “verticals”, such as IoT:
"Massive MIMO channel-aware decision fusion." IEEE Transactions on Signal Processing 63.3 (2014): 604-619.
"Massive MIMO for wireless sensing with a coherent multiple access channel." IEEE transactions on signal processing 63.12 (2015): 3005-3017.
"Massive MIMO for decentralized estimation of a correlated source." IEEE Transactions on Signal Processing 64.10 (2016): 2499-2512.
"Analyzing random access collisions in massive IoT networks." IEEE Transactions on Wireless Communications 17.10 (2018): 6853-6870.
"Random access analysis for massive IoT networks under a new spatio-temporal model: A stochastic geometry approach." IEEE Transactions on Communications 66.11 (2018): 5788-5803.
4) I would like the authors to somewhat rephrase/strengthen the central/final part of Sec. I, by (i) adding a summarizing paragraph at the end of the literature review which highlights the current limitations/shortcomings of existing pilot decontamination methods and (ii) rephrasing the statement of contributions in a more effective and structured fashion (e.g. with a bullet list).
5) Please add both notation and organization paragraphs at the end of Sec. I.
6) In the system model description, the authors consider (i) Rayleigh fading channels and (ii) a 3 hexagonal cells. I would like them to clarify whether these assumptions are restricting (i.e. constitute a limitation of the proposed approach).
7) For the sake of a complete assessment, I would like the authors to discuss the computational complexity involved in both Algorithms 1 and 2.
8) In Figs. 10, 11 and 13, please add the corresponding reference (in the legend item) where each baseline is taken from.
9) In conclusion section, the paragraph on future directions of research should be improved more.
10) Please double-check the whole paper for possible typos and examples of English misuse, e.g.:
“the number of cell users isK “ -> “the number of cell users is K”
“from Ω1toΩkc” -> “from Ω1 to Ωkc”
“Figure 12 shows the effect of separation parameterλ” -> “Figure 12 shows the effect of separation parameter λ”
Author Response
Response to Reviewer 2 Comments
Overall, this is a nice contribution on the topic of pilot decontamination in massive MIMO systems. Still, before publication, it is my opinion that the following major comments should be addressed by the authors:
- Please define all the acronyms in their first use, e.g. TDD.
Answer: As instructed by the esteemed reviewer, all the acronyms in their first use are defined throughout the manuscript, indicated by red color in the revised manuscript.
- I – “Various researchers have proposed different approaches and mitigated the issue of PC, which are shown in Figure 1” -> “Various researchers have proposed different approaches to mitigate the issue of PC, as categorized via Figure 1” When introducing the benefits of massive MIMO technology (at the beginning of Sec. I), the authors may also want to highlight its benefit to some important “verticals”, such as IoT:
"Massive MIMO channel-aware decision fusion." IEEE Transactions on Signal Processing 63.3 (2014): 604-619.
"Massive MIMO for wireless sensing with a coherent multiple access channel." IEEE transactions on signal processing 63.12 (2015): 3005-3017.
"Massive MIMO for decentralized estimation of a correlated source." IEEE Transactions on Signal Processing 64.10 (2016): 2499-2512.
"Analyzing random access collisions in massive IoT networks." IEEE Transactions on Wireless Communications 17.10 (2018): 6853-6870.
"Random access analysis for massive IoT networks under a new Spatio-temporal model: A stochastic geometry approach." IEEE Transactions on Communications 66.11 (2018): 5788-5803.
Answer: In the introduction section of the manuscript, we highlighted some prominent benefits of the massive MIMO systems with respect to the Massive IoT applications and cited some suggested related research papers by reviewers in the manuscript.
- I would like the authors to somewhat rephrase/strengthen the central/final part of Sec. I, by (i) adding a summarizing paragraph at the end of the literature review which highlights the current limitations/shortcomings of existing pilot decontamination methods and (ii) rephrasing the statement of contributions in a more effective and structured fashion (e.g. with a bullet list).
Answer:
- The following mentioned paragraph is added at the end of related work section (section 2) of the manuscript, highlighting the limitations/shortcomings in the existing research work regarding the pilot decontamination.
“Literature review indicated that various pilot decontamination schemes (i.e. pilot design, channel estimation, pre-coding, and pilot allocation schemes), are limited due to the increase in pilot overhead. Further, these schemes employ the power control method thereby limiting the performance of these systems when there is an increase in the number of antennas. In addition, many proposed schemes exhibited computational complexity. Therefore, we focused on these issues and proposed FPR and AFPS schemes, which significantly mitigated the pilot overhead and inter-cell interference (ICI), resulting in a significant reduction in the PC issue, and the computation complexity, in terms of improved sum rates”.
- As instructed by esteemed reviewers, the main contributions are highlighted in the manuscript, in the introduction section, and presented in bullet points.
- Please add both notation and organization paragraphs at the end of Sec. I.
Answer:
The following table representing notations has been included as Appendix A at the end of the manuscript.
|
Notations |
Descriptions |
|
(.)H |
Hermitian transpose |
|
||.|| |
two-norm |
|
IM |
size-M Identity matrix |
|
(.)* |
Conjugate value |
|
|.| |
norm |
|
CN |
circularly-symmetric Gaussian distribution |
|
bold font upper case letters |
matrices |
|
bold font lower case letters |
vectors |
Further, the following paragraph is included in the introduction section describing the organization of this manuscript.
“The remainder of the manuscript is organized as: Section 2 describes related work. In section 3, the system model of the proposed FPR scheme is presented in detail, further, the capacity of central and edge users has been derived. In section 4 the proposed AFPS based on user grouping, is presented. The performance of our proposed schemes is compared with previous methods via simulations in section 5. Finally, the conclusions are described in Section 6.
- In the system model description, the authors consider (i) Rayleigh fading channels and (ii) a 3 hexagonal cells. I would like them to clarify whether these assumptions are restricting
Answer:
- Employing a Rayleigh Fading channel is not a restriction. However, in our work, we assumed a Rayleigh fading channel because of the fact that the Rayleigh fading channels have a positive influence on system performance [9] i.e. while using Monte Carlo Simulations in Rayleigh fading channels, the system performance can be improved by achieving better estimates.
- In an infinite plane, a hexagonal cell structure is composed of base stations (BSs) transmitting signals to mobile users at deterministic locations, which are uniformly distributed over the individual BS cell area. There is only one sector per cell for convenience, and the BSs are located at the center of each cell with the inner radius d. Each consumer is connected to the nearest BS. Hence, we only took 03 multiplexing cells from that field, in a given situation.
Hexagonal cells are used in our work because these are used for network planning and analysis, in order to model the macro BSs [10], [11, 12]. Additionally, in our proposed work the users at the edge of the hexagonal cells are more prone to interference. Therefore, higher spectral efficiency can be achieved, while employing a suitable pilot allocation scheme in a well-planned cellular networks.
- Wu, Q. and Q. Liang, Increasing the Capacity of Cellular Network With Nested Deployed Cooperative Base Stations. IEEE Access, 2018. 6: p. 35568-35577.
- Garg, V., Wireless Communications & Networking. 2007.
- Saquib, N., et al., Interference management in OFDMA femtocell networks: issues and approaches. IEEE Wireless Communications, 2012. 19(3): p. 86-95.
- Yang, X. and A. Fapojuwo, Performance analysis of hexagonal cellular networks in fading channels. Wireless Communications and Mobile Computing, 2015. 16.

Round 2
Reviewer 2 Report
Overall, this is a nice contribution on the topic of pilot decontamination in massive MIMO systems.
Additionally, the authors have satisfactorily addressed my previous comments and modified their manuscript accordingly. Hence, I am glad to recommend the present work for publication.